# IgG4-Related Disease of the Oral Cavity. Case Series from a Large Single-Center Cohort of Italian Patients

**DOI:** 10.3390/ijerph17218179

**Published:** 2020-11-05

**Authors:** Andrea Rampi, Marco Lanzillotta, Gaia Mancuso, Alessandro Vinciguerra, Lorenzo Dagna

**Affiliations:** 1Otorhinolaryngology Unit, Division of Head and Neck Department, IRCCS San Raffaele Scientific Institute, 20132 Milan, Italy; rampi.andrea@hsr.it (A.R.); vinciguerra.alessandro@hsr.it (A.V.); 2School of Medicine, Vita-Salute San Raffaele University, 20132 Milan, Italy; mancuso.gaia@hsr.it (G.M.); dagna.lorenzo@unisr.it (L.D.); 3Division of Immunology, Rheumatology, Allergy and Rare Disease (UnIRAR), IRCCS San Raffaele Scientific Institute, 20132 Milan, Italy

**Keywords:** IgG4-related disease, oral lesions, differential diagnosis, histology, biopsy

## Abstract

A series of destructive and tumefactive lesions of the oral cavity are increasingly recognized as part of the IgG4-related disease (IgG4-RD) spectrum. We herein examined the clinical, serological, radiological, and histological features of a series of patients referred to our clinic because of oral cavity lesions ultimately attributed to IgG4-RD. In particular, we studied 6 consecutive patients out of 200 patients referred to the immunology outpatient unit who presented with erosive and/or tumefactive lesions of the oral cavity. All patients underwent serum IgG4 measurement, nasal endoscopy, radiological studies, and histological evaluation of tissue specimens. The histological studies included immunostaining studies to assess the number of IgG4+ plasma cells/High-Power Field (HPF) for calculation of the IgG4+/IgG+ plasma cell ratio. Six patients (3% of the entire cohort) were diagnosed with IgG4-RD of the oral cavity based on histological evaluation. A major complaint at presentation was oral discomfort due to bulging mass. A mild to no increase in serum IgG4 was observed. Different patterns of organ involvement were associated with oral lesions. Five patients were treated with immunosuppressive therapy and two patients promptly responded to B-cell depletion with rituximab. Watchful waiting was decided in one patient with no major clinical symptoms. Involvement of the oral cavity is an infrequent manifestation of IgG4-RD but should be taken into consideration as a possible differential diagnosis of tumefactive or erosive lesions once neoplastic conditions are excluded. A histological examination of biopsy samples from the oral cavity represents the mainstay for diagnosis of IgG4-RD.

## 1. Introduction

IgG4-related disease (IgG4-RD) is a systemic immune-mediated condition potentially affecting any organ, characterized by tissue inflammation and fibrotic outcome [1,2]. It can present with both insidious or acute manifestations and lead to end-stage organ damage if left untreated [3]. IgG4-RD encompasses a spectrum of clinical manifestations considered as independent disorders for decades, such as type I autoimmune pancreatitis, retroperitoneal fibrosis, Riedel’s thyroiditis, Mikulicz’s disease, and hypertrophic pachymeningitis [1]. Awareness of the multi-faceted presentation of IgG4-RD rapidly grew worldwide after its recognition as a distinct entity in 2003, and diagnosis is now confidently achieved across different medical specialties [1,4,5].

In particular, four clinical phenotypes of IgG4-RD have been defined based on organ involvement: (i) pancreaticobiliary disease; (ii) retroperitoneal fibrosis with or without aortitis; (iii) head- and neck-limited disease; (iv) Mikulicz’s syndrome with systemic involvement [1,6,7,8]. While pancreatic, retroperitoneal, salivary and lacrimal gland involvement historically represent prototypical manifestations of IgG4-RD, also being considered among the inclusion criteria for IgG4-RD classification, lesions in the head and neck are less specific and well-characterized, thus making differential diagnosis of these cases more challenging [4,9]. IgG4-RD in the head and neck region can, indeed, affect a variety of anatomical structures including the orbits, meninges, ears, thyroid, skull bones, sinuses, and oral cavity, mimicking a number of infectious and neoplastic conditions [1,4,10]. Of note, while nasal involvement from IgG4-RD has been consistently described, lesions affecting the oral cavity have been infrequently reported, possibly representing an overlooked manifestation of IgG4-RD [11,12,13,14]. Moreover, failure to recognize IgG4-related oral lesions might lead to unnecessary disruptive surgery, greatly affecting patients’ quality of life.

In the present work, we report on six patients with atypical localizations of IgG4-RD in their oral cavities and describe the main clinical, pathological, serological, and radiological findings of a unique presentation that might complicate the differential diagnostic process.

## 2. Methods

Six consecutive patients out of 200 patients, referred to the immunology outpatient unit with a diagnosis of IgG4-RD involving the oral cavity evaluated at San Raffaele Scientific Institute between January 2015 and February 2020, were included in this retrospective study. All the patients fulfilled the available IgG4-RD diagnostic and classification criteria [2,6]. Demographic variables, including age and sex, as well as radiological and histopathological parameters were collected for all patients. The histological studies included immunostaining studies to assess the number of IgG4+ plasma cells/High-Power Field (HPF) for calculation of the IgG4+/IgG+ plasma cell ratio and to rule out immunoglobulin light chain restriction.

The present study has been approved by the San Raffaele Hospital ethics committee (approved on 15/7/2020) in adherence with the Declaration of Helsinki.

M.L., G.M. and L.D. performed clinical evaluations and gathered laboratory data of the patients, while A.R. and A.V. performed the nasal endoscopy and tissue biopsy.

## 3. Case Presentation

Case 1: A 35-year-old female was admitted in April 2015 for a rapidly progressive frontal headache and complete visual loss in the right eye. Imaging studies of the head and neck revealed diffuse pachymeningitis with bone erosions, thickening of the nasal septum, hard palate tumefaction, and bilateral optic neuritis (Figure 1a). Her medical history was otherwise unremarkable. In January 2014, she first experienced a foreign body sensation in her mouth and noticed a bulging of her hard palate. A first biopsy for histological examination showed a diffuse sclerotic tissue with a non-specific inflammatory infiltrate. A tentative diagnosis of vasculitis was given and the patient was treated with a 3-month course of steroid therapy leading to clinical improvement. At admission, the hard palate lesion recurred and a new biopsy showed storiform fibrosis with a IgG4/IgG-positive plasma cell ratio >40% and 100 IgG4-positive plasma cells/High-Power Field (HPF) (Figure 1e,f). Serum IgG4 was 151 mg/dL (normal 10–140 mg/dL). A definite diagnosis of IgG4-RD was made, and the patient was treated with pulsed steroid therapy (1 gm/day for 3 days) because of her acute neurological symptoms. Oral prednisone (0.6 mg/kg/day) was then initiated and gradually tapered over 6 months with a resolution of the palate lesion, partial amelioration of the headache, but only minor improvement of visual acuity. In December 2015, the patient experienced a seizure and a relapse of the hard palate lesion (Figure 1c), consistent with an IgG4-RD flare in the meninges and oral cavity. Two 1-g doses of rituximab 15 days apart were infused as second-line therapy. Two months later, a clinical and radiological examination showed marked improvement of the hypertrophic pachymeningitis and resolution of the hard palate lesion (Figure 1b,d).

Case 2: A 20-year-old female was referred in June 2012 for a 4-month history of cervical lymphadenopathy and bilateral parotid glands swelling. Oroscopy also unveiled an asymptomatic involvement of the pterygopalatine fossa and oropharyngeal structures, with clear bulging of the hard palate (Figure 2). An excisional biopsy of a cervical lymph node was diagnostic for IgG4-RD. Other infectious and neoplastic conditions were excluded by a thorough serological and hematological workup. Serum IgG4 was three times the upper limit of normal (421 mg/dl, normal 10–140 mg/dL) thus supporting a diagnosis of Mikulicz’s syndrome with lymph node and palate involvement. Rituximab was preferred over glucocorticoids and administered with four 375 mg/m^2^ weekly intravenous infusions leading to a prompt resolution of the parotid swelling and palate mass at 4-month follow-up.

Case 3: A 45-year-old male was referred in November 2019 for a 9-month history of mild dyspnea (on exertion), cough, nasal obstruction and bulging of his left nasolabial fold. An 18-fluorodeoxyglucose (18-FDG) positron emission computed tomography (PET-CT) scan showed bilateral ground-glass opacities, an intranasal mass with downstream extension to the superior alveolar processes, and disease activity in cervical and hilar lymph nodes. A biopsy of the nasolabial tissue showed an inflammatory pseudotumor with storiform fibrosis, more than 80 IgG4-positive plasma cells/HPF, and a IgG4/IgG-positive plasma cell ratio of 30–40%, a result that was highly suggestive of IgG4-RD. Serum IgG4 was 420 mg/dl (normal 10–140 mg/dL). Mimicker neoplastic, inflammatory, and infectious disorders were ruled out through an extensive serological and pathological evaluation. Corticosteroid therapy was started at a daily prednisone dose of 0.6 mg/kg followed by gradual tapering over six months. Four months after initiation of the steroid therapy, a resolution of the cough and dyspnea were observed, but the bulging of the nasolabial fold was clinically stable. Because of the poor response of IgG4-RD to steroids in the oral cavity, surgical debulking is currently under consideration.

Case 4: A 37-year-old female was referred in December 2017 for gradual onset of a dry cough, sore throat, and bilateral cervical lymphadenopathy. At oroscopy, a peritonsillar abscess was observed together with a hard mass on the left side of the soft palate close to the uvula (Figure 3a). The abscess was successfully treated with antibiotics, but the palate mass remained stable as did tonsillar swelling. A biopsy of the tonsillar tissue was performed, and important fibrosis was observed during surgery similar to the sequelae of local radiotherapy. Histology showed storiform fibrosis, 180 IgG4-positive plasma cells/HPF, and an IgG4/IgG-positive plasma cells ratio of 30%, which is highly suggestive of IgG4-RD. Serum IgG4 was 131 mg/dL (normal 10–140 mg/dL). An operative diagnosis of IgG4-RD was made and 0.6 mg/kg/day of oral methylprednisolone was introduced and gradually tapered over 6 months. Four months later the tonsillar swelling and palate mass resolved (Figure 3b).

Case 5: A 57-year-old male was referred for melena in September 2018. Radiological examination disclosed fibrotic tissue in the retroperitoneum surrounding the abdominal aorta and encasing the small bowel (Figure 4a). A whole-body 18-FDG PET-CT scan also showed a mass-forming lesion at the base of the tongue extending to the posterior wall of the oropharynx (Figure 4b). A biopsy of the lesion in the oral cavity was not diagnostic. A biopsy of the retroperitoneal tissue showed fibrosis with IgG4/IgG > 40% and IgG4/HPF > 100, thus being diagnostic of IgG4-RD. Serum IgG4 levels were normal as oftentimes occurs in the retroperitoneal involvement of IgG4-RD [1]. The patient was started on 0.6 mg/kg of daily prednisone for one month in March 2019. Steroid treatment was tapered over 6 months to low prednisone (5 mg daily) and maintained for 12 months. After 6 months, a follow-up radiological evaluation showed a marked improvement of both lesions in the oral cavity and retroperitoneal fibrosis.

Case 6: A 29-year-old male was referred in April 2013 for bilateral swelling of parotid and submandibular glands with discomfort when chewing. A PET-CT scan showed symmetric 18-FDG uptake at the salivary glands and identified an asymptomatic involvement of the rhinopharynx and tongue base (Figure 5). At oroscopy, a hypertrophy of the lymphoid tissue at the base of the tongue was observed. Serum IgG4 was 257 mg/dl (normal 10–140 mg/dL), reinforcing a diagnosis of IgG4-related Mikulicz’s disease with oral cavity involvement. A serological test for autoantibodies and angiotensin-converting enzyme was normal, thus excluding mimicker conditions such as Sjogren syndrome and sarcoidosis. Repeated biopsies of the labial salivary glands and of the oropharyngeal tissue yielded non-specific results. Because the patient was largely asymptomatic and concerned about steroid treatment, watchful waiting was considered. Subsequent radiological evaluations up to present have shown a stable disease.

The mean age of the patients ws 37.7 years (SD 12.8 years). The main features of the six patients in the sample are presented in Table 1 and Table 2.

## 4. Discussion

Prior to 2003, different illnesses such as Mikulicz’s syndrome, autoimmune pancreatitis, hypertrophic pachymeningitis, and Riedel’s thyroiditis were all considered separate entities, and only 17 years ago they were recognized as part of the unified spectrum of IgG4-related disorders [1,15,16,17,18]. Since then, the literature on IgG4-RD has grown exponentially, and manifestations in nearly any organ have been attributed to this novel fibro-inflammatory entity [1,4,5,19]. Yet, few cases of IgG4-related lesions of the oral cavity have been reported, suggesting that this specific organ involvement might be unusually observed [12,13,14]. A comprehensive description of IgG4-RD manifestations in the oral cavity is, indeed, currently lacking and it is possible that this involvement might be underdiagnosed.

In our large single-center cohort, localizations of IgG4-RD in the oral cavity were observed in 6 of 200 patients (3%), a frequency similar to pleural, sinonasal, and pharyngeal involvement [1]. The hard palate seems to be a preferential site of IgG4-RD involvement (three cases) followed by the lymphoid tissue at the tongue base (two cases) and by the alveolar processes (one case). In all cases, IgG4-RD localization in the oral cavity was largely asymptomatic and presented with a hard, non-tender mass growing insidiously and covered by erythematous mucosa. When biopsied, tissue fibrosis was described in all cases as well as a variable amount of IgG4-positive lymphoplasmacytic infiltrate. Interestingly, in contrast to other series, none of our patients except one (Case 1) showed erosive lesions of the midline structures, possibly indicating that IgG4-RD in the oral cavity might follow different pathological stages. Previous reports, in fact, describe insidious erosion of midline facial structures occurring over 2–3 years in the absence of other systemic IgG4-RD manifestations, with patients complaining of symptoms directly caused by tissue loss in the hard palate such as chronic sore throat, difficulty swallowing (nasal regurgitation), and progressive nasalization of speech [4,20,21]. Conversely, in the present case series, oral cavity involvement was diagnosed in the context of systemic organ manifestations of IgG4-RD at the time of disease onset. Moreover, a patient assessment with 18-fluorodeoxyglucose PET-CT at baseline likely allowed early recognition and treatment of palate involvement before the instauration of palate perforation [1,11].

IgG4-RD of the oral cavity, especially if isolated and not associated with other suggestive organ manifestations, can be challenging to recognize for several reasons. First, according to our experience, findings from physical examinations are largely non-specific, with patients presenting a swelling of mucosal and submucosal tissues with local hyperemia. However, although we could not identify any characteristic clinical feature, the absence of ulcers, tissue necrosis, fever, mucopurulent drainage, and local pain were helpful in orienting the diagnosis towards an inflammatory rather than infectious or neoplastic condition [4,22,23,24,25,26,27,28,29,30,31,32,33,34,35,36,37,38]. The prompt response to steroid therapy also reinforced this consideration. Moreover, a histological examination was required to achieve a final diagnosis and to rule out vasculitis or granulomatous conditions. Second, serological findings also have relevant shortcomings as ESR and CRP are typically not of use when supporting a diagnosis of IgG4-RD [1,19,39,40]. Additionally, serum IgG4 might support a clinical suspicion of IgG4-RD only if it is markedly elevated, as mild to no elevation can be observed in a number of mimicker conditions [1,9]. Finally, due to the unusual description of this presentation, radiologic findings of IgG4-RD affecting the oral cavity have not been thoroughly studied and validated, thus offering no additional help in the differential diagnosis. In the cases presented, both CT and MRI showed contrast-enhanced focal mass lesions while bone erosion was observed only in one case of palatal localization [41].

In light of the abovementioned considerations, our experience suggests that diagnosis of IgG4-RD in the oral cavity should rely on a combination of clinical, serological, radiological, and histopathological findings because none of these approaches alone has sufficient accuracy as in other IgG4-RD manifestations [19]. Of note, pathological examination might also have relevant shortcomings because, as opposed to more typical organ involvement, the oral cavity is not among the anatomical sites included in the “Consensus statement on the Pathology of IgG4-RD”, the reference guideline document for histological diagnosis of IgG4-RD [42]. Histological findings in the oral cavity, in fact, have not been validated and hallmark features of IgG4-RD, such as storiform fibrosis, obliterative phlebitis, ectopic lymphoid structures, and IgG4 infiltrate, might not be as commonly observed as in other anatomical sites [1]. From a clinical standpoint, the concomitant involvement of typically affected organs such as the retroperitoneum and salivary glands represented an important clue to consider mass-forming lesions in the oral cavity as part of a multi-organ presentation of IgG4-RD. Of note, while serum IgG4 is typically normal in localized head and neck manifestations of IgG4-RD, patient no. 1, no. 2, no. 3 and no. 6 showed mild to moderately elevated levels, probably in the context of a systemic multi-organ disease. Additionally, serum IgG4 concentrations have shown a relevant shortcoming for diagnostic purposes in different international studies and should not be used alone to orient the diagnosis [43,44,45].

Oral lesions due to IgG4-RD responded well to glucocorticoid therapy in most cases with prompt shrinkage of the fibro-inflammatory mass [46,47]. B-cell depletion with rituximab also led to satisfactory improvement of IgG4-RD in the oral cavity when used as first- and second-line therapy in patient no. 1 and no. 2. In patient no. 6, watchful waiting was decided due to asymptomatic disease with no further disease progression at the longest available follow-up. In patient no. 3, in contrast, steady response of lung IgG4-RD to glucocorticoids was not paralleled by a satisfactory response of the localizations at the alveolar processes. At present, it is difficult to explain the different outcomes of these two distant IgG4-RD involvements treated with the same corticosteroid regimen, although we might speculate that the oral cavity lesion was more fibrotic and, thus, less responsive to immunosuppressive therapy. Surgical debulking in these cases could represent a valid therapeutic strategy to relieve local symptoms and to obtain an informative tissue sample. Little is, indeed, known about the pathogenesis of IgG4-RD and its oral cavity manifestations, thus limiting the medical therapeutic armamentarium for this rare disorder to steroids and B cell-depleting agents. Experimental evidence suggests a cooperation of B and T cells in orchestrating an antigen-specific immune response ultimately leading to a fibrotic outcome, but more specific therapeutic targets are yet to be defined [1,43,46,48,49,50].

## 5. Conclusions

The oral cavity is a rare but probably underestimated localization of IgG4-RD, which lacks a single reliable instrument for its diagnosis. Nevertheless, even when it presents with a single non-specific localization as in the oral cavity, IgG4-RD is a systemic disease, and as such it must be investigated. Therefore, the most adequate strategy to manage an oral lesion suspected for IgG4-RD manifestation should include clinical, serological, radiological, and histopathological studies assessed by different specialists, and only the combination of these findings may lead to a reliable diagnosis. The experience in oral lesions is too limited to find a targeted therapy, which is then based on the suggested treatments for other sites, basically steroids and B cell-depleting agents. Possible future developments would be the discovery of new drugs and the definition of more specific therapeutic algorithms, if not for any sites involved, at least for the four phenotypes identified.

## Figures and Tables

**Figure 1 ijerph-17-08179-f001:**
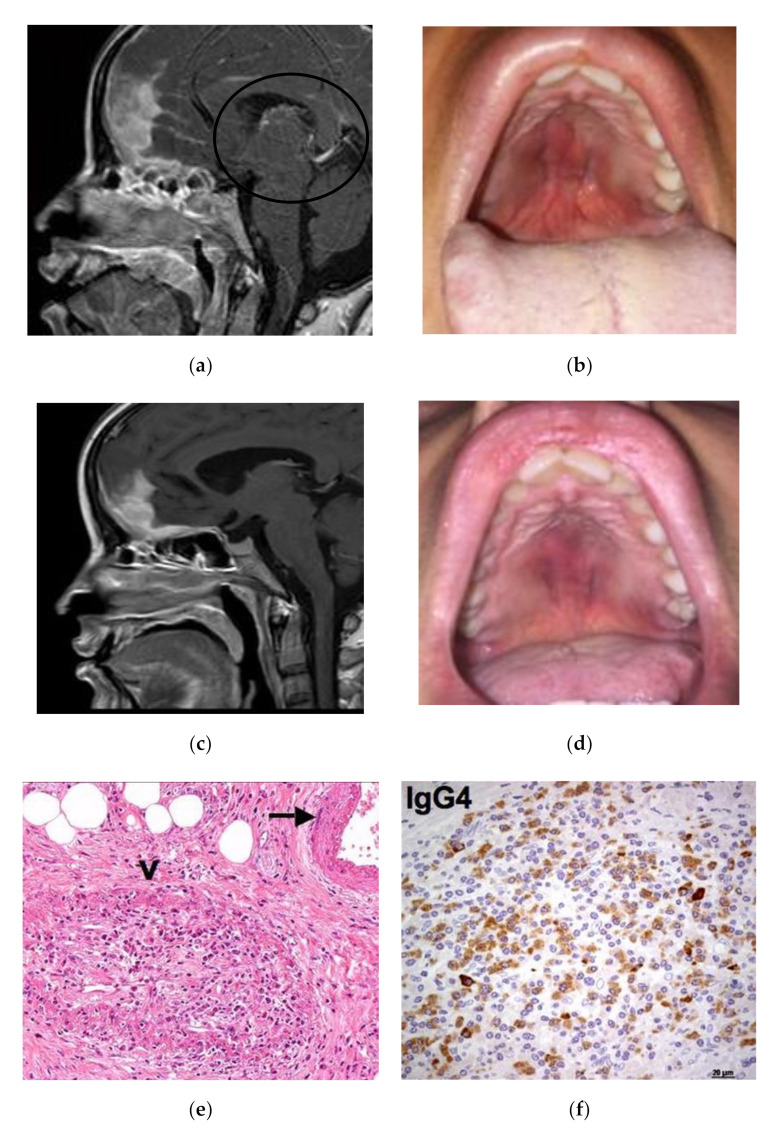
Patient 1: (**a**) Sagittal Magnetic Resonance Imaging (MRI ) view showing involvement of nasal septum, hard palate and pachymeninges (black circle). (**b**) The relapse of the hard palatal lesion. (**c**) Radiological and (**d**) clinical remission after rituximab therapy. (**e**) Histologic examination of the hard palatal lesion showing storiform fibrosis, plasma cell infiltrate and obliterative phlebitis (arrow). (**f**) Immunohistochemistry study evaluating IgG4+ cells.

**Figure 2 ijerph-17-08179-f002:**
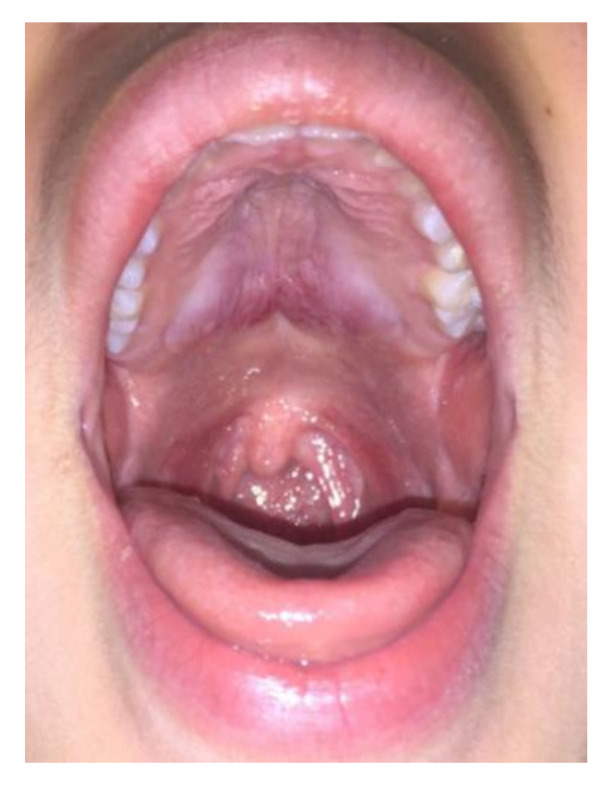
Patient 2: hard palatal lesion.

**Figure 3 ijerph-17-08179-f003:**
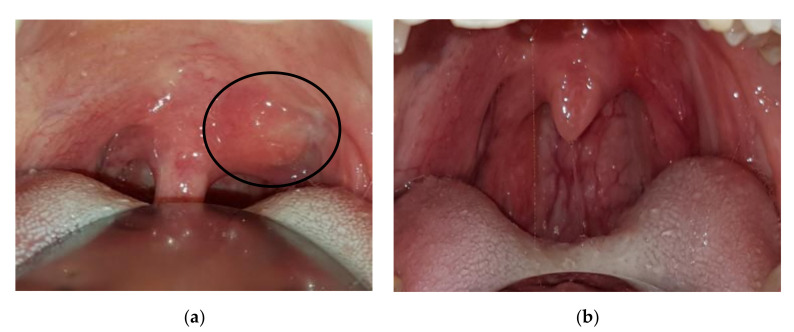
Patient 4: (**a**) Periuvular lesion of the soft palate (circle), (**b**) resolution after corticosteroid treatment.

**Figure 4 ijerph-17-08179-f004:**
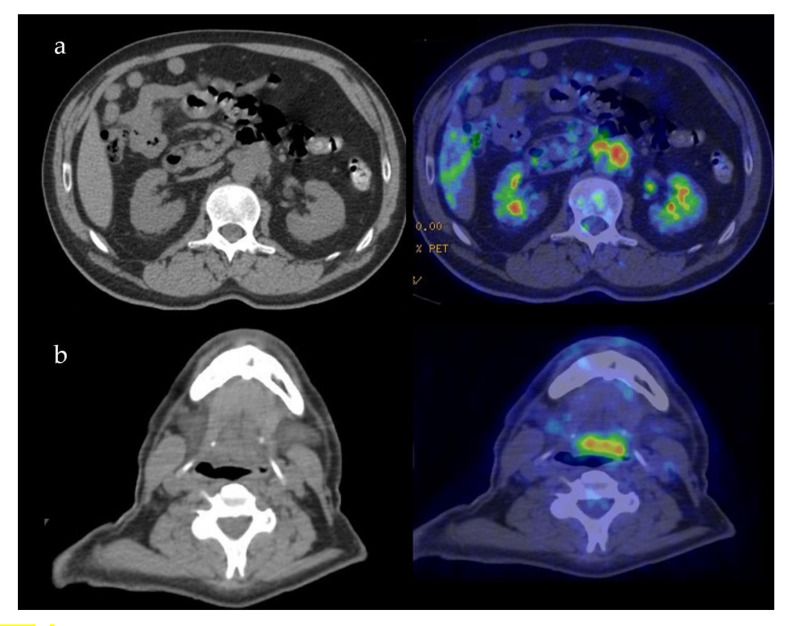
Patient 5: PET-CT findings of the retroperitoneum (**a**) and oropharynx (**b**).

**Figure 5 ijerph-17-08179-f005:**
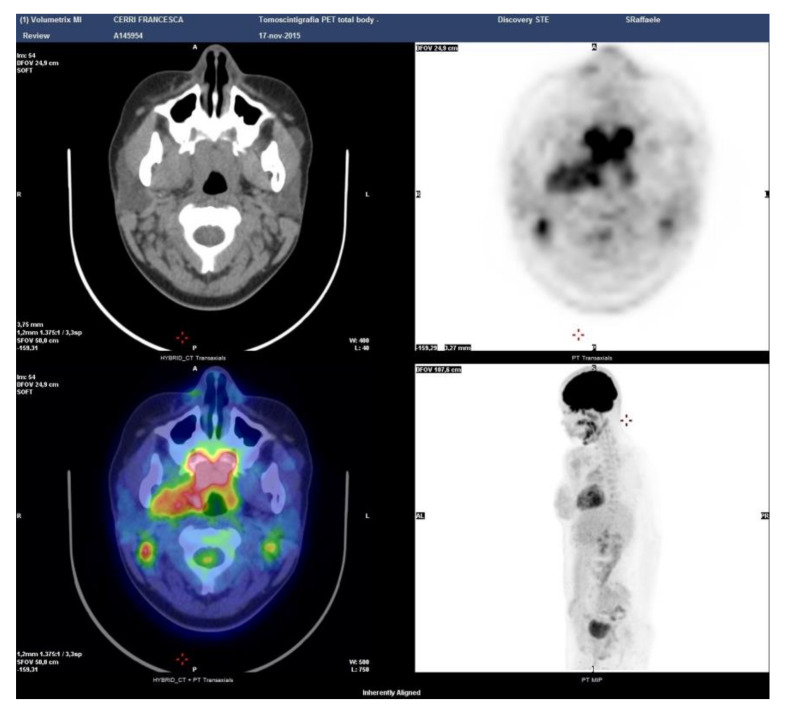
Patient 6: PET-CT scan showing the sites of disease activity.

**Table 1 ijerph-17-08179-t001:** Main demographic features, sites of lesion and serum findings.

Patient	Age, Sex	Oral Cavity Involvement	Other Structures Involved	ESR ^1,2^	CPR ^1,3^	Serum IgG4 ^1,4^
1	35, F	Hard palate	Pachymeninges, optic nerves, nasal septum	33	44	151
2	20, F	Hard palate	Cervical lymph nodes, parotids, oropharynx, pterygopalatine fossa	5	0.3	421
3	45, M	Superior alveolar processes	Nasal and maxillary structures, cervical lymph nodes, lungs			420
4	37, F	Tonsillar and peritonsillar region	Cervical Lymph nodes	72	8	131
5	57, M	Tongue base extending to oropharynx	Retroperitoneum encasing aorta	28	1.3	30
6	29, M	Tongue base	Salivary glands, Rhino pharynx	8	23	257

^1^ at diagnosis; ^2^ Erythrocyte sedimentation rate, mm/h, Normal values (n.m.) 1–20; ^3^ C-reactive protein, mg/L n.m. <6 ^4^ mg/L, n.m 10–140 mg/dL

**Table 2 ijerph-17-08179-t002:** Histologic findings, therapies and outcomes.

Patient	Histologic Findings	First-Line Therapy and Outcome	Second-Line Therapy and Outcome
1	Palate: storiform fibrosis, IgG4+: 100 × HPF, IgG4+/IgG+ plasma cells > 40%	Prednisone 1 gm/die for 3 days, then 0.6 mg/kg/die and gradual tapering. Partial response.	Two 1 gm doses of rituximab 15 days apart. Marked improvement.
2	Small salivary gland: fibrosis, IgG4+ 50 × HPF, IgG4+/IgG+ plasma cells 70%	Rituximab 375 mg/m^2,^ four weekly infusions IV. Prompt resolution of symptoms.	
3	Nose: storiform fibrosis, IgG4+: > 80 × HPF, IgG4+/IgG+ plasma cells 30–40%	Prednisone 0.6 mg/kg/die and gradual tapering. Lung response, persistence of maxillofacial lesions.	Under consideration for surgery.
4	Tonsil: fibrosis, IgG4+ > 180 × HPF, IgG4+/IgG+ plasma cells 30%	Prednisone 0.6 mg/kg/die and gradual tapering. Complete response.	
5	Tongue: undiagnostic. Retroperitoneum: IgG4+: > 100 × HPF, IgG4+/IgG+ plasma cells > 40%	Prednisone 0.6 mg/kg/die and gradual tapering. Marked improvement.	
6	Labial salivary glands and tongue base: unspecific inflammation	Watchful waiting. Stable disease.

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
