# Peer review of "IgG4-Related Disease of the Oral Cavity. Case Series from a Large Single-Center Cohort of Italian Patients"

_ijerph, 2020, doi:10.3390/ijerph17218179_

Round 1

Reviewer 1 Report

IgG4-related disease of the oral cavity. Case series from a large single-center cohort of Italian patients

This is an interesting study on 6 consecutive patients out of 200 patients affected by IgG4 related disease, who presented with erosive and/or tumefactive lesions of the oral cavity. This presentation is indeed rather unusual.

I suggest to change the histological picture since it does not show a storiform pattern and to provide also an immunohistochemical image with immunohistochemical labelling.

In addition the authors should state if an immunohistochemistry panel was used to rule out a lymphoma. 

Author Response

Reviewer 1

This is an interesting study on 6 consecutive patients out of 200 patients affected by IgG4 related disease, who presented with erosive and/or tumefactive lesions of the oral cavity. This presentation is indeed rather unusual.

We thank the reviewer for appreciating our work.

I suggest to change the histological picture since it does not show a storiform pattern and to provide also an immunohistochemical image with immunohistochemical labelling.

We thank the reviewer for the suggestion. We changed the histological pictures accordingly.

In addition the authors should state if an immunohistochemistry panel was used to rule out a lymphoma.

We thank the reviewer for his/her insight. We added this information in the caption.

Reviewer 2 Report

The topic of this manuscript may be clinically interesting and useful for readers in a specific area of the journal. However, the paper presents some lacks and weaknesses to be revised, as described below.

In the Abstract, as well as throughout the manuscript, the personal forms i.e. “to our clinic” (..) “we studied 6 consecutive patients out of 200 patients of our cohort” should be avoided.

In the Introduction:

- more information should be provided on the rationale of the study, better focusing on the differences with the previous publications;

- the following statement “The present study has been approved by the San Raffaele Hospital ethics committee in adherence with the Declaration of Helsinki” should be moved in a specific paragraph before the Case presentations, adding also the following information:

- the period of consecutive sample recruitment;

- the information on the total number of patients from which the cases were selected (provided only in the abstract);

- more information on the specific criteria and modalities used to recruit the subjects in the study;

- the demographics, as mean age with standard deviation of the six patients then described into details;

- initials and qualifications of the operators performing each step of the study.

Figures

The intraoral figures should be of higher quality for a scientific publication.

The References’ list presents some orthographic errors, the citations in the text does not always follow the journal guidelines.

Editing of English language and style (i.e. punctuations, spaces) from a native speaker may be useful.

Author Response

Reviewer 2

The topic of this manuscript may be clinically interesting and useful for readers in a specific area of the journal. However, the paper presents some lacks and weaknesses to be revised, as described below.

In the Abstract, as well as throughout the manuscript, the personal forms i.e. “to our clinic” (..) “we studied 6 consecutive patients out of 200 patients of our cohort” should be avoided.

We thank the reviewer for the suggestion, we changed the text accordingly.

In the Introduction:

- more information should be provided on the rationale of the study, better focusing on the differences with the previous publications;

We thank the reviewer for the insight. We stressed out how there is a paucity of literature related to IgG4-RD oral lesions compared to nasal ones.

- the following statement “The present study has been approved by the San Raffaele Hospital ethics committee in adherence with the Declaration of Helsinki” should be moved in a specific paragraph before the Case presentations, adding also the following information:

- the period of consecutive sample recruitment;

- the information on the total number of patients from which the cases were selected (provided only in the abstract);

- more information on the specific criteria and modalities used to recruit the subjects in the study;

- the demographics, as mean age with standard deviation of the six patients then described into details;

- initials and qualifications of the operators performing each step of the study.

We thank the reviewer for the suggestions. We created and implemented the methods section with the above suggested information.

Figures

The intraoral figures should be of higher quality for a scientific publication.

We thank the reviewer for pointing this out. We updated the figure following the journal’s guidelines.

The References’ list presents some orthographic errors, the citations in the text does not always follow the journal guidelines.

 We thank the reviewer for the comment. We fixed the mistakes in the references’ list.

Editing of English language and style (i.e. punctuations, spaces) from a native speaker may be useful.

We thank the reviewer for the comment. We edited the style and language.

Reviewer 3 Report

This manuscript addresses an interesting topic and presents six cases of IgG4-related disease of the oral cavity. The authors emphasised that the diagnosis of IgG4-RD in the oral cavity should be based on a combination of clinical, serological, radiological, and histopathological findings. Overall, the manuscript reads well, the cases are well presented, and the discussion is well-argued and supported by references.
However, some minor points could be addressed to enhance the readability of the manuscript.
- L 67-68: Please provide the number of ethical registration if possible.
- It would be useful to have some clinical photos or histological sections for Case 3 and Case 5.
- L 234 -249: refer to the patients with their case numbers.

Author Response

Reviewer 3

This manuscript addresses an interesting topic and presents six cases of IgG4-related disease of the oral cavity. The authors emphasised that the diagnosis of IgG4-RD in the oral cavity should be based on a combination of clinical, serological, radiological, and histopathological findings. Overall, the manuscript reads well, the cases are well presented, and the discussion is well-argued and supported by references.

We deeply thank the reviewer for supporting our work.

However, some minor points could be addressed to enhance the readability of the manuscript.

- L 67-68: Please provide the number of ethical registration if possible.

We thank the reviewer for pointing this out. The ethical registration number is not available, but we have the date when the approval was granted, that has been added in the manuscript.

- It would be useful to have some clinical photos or histological sections for Case 3 and Case 5.

We thank the reviewer for the suggestion. We added the imaging findings of patient 3, while unfortunately clinical and histological images were not available.

- L 234 -249: refer to the patients with their case numbers.

We thank the reviewer for the useful suggestion. We changed the manuscript accordingly.

Round 2

Reviewer 2 Report

Dear Authors,

thank you for your corrections on the manuscript

and good luck for the publication.

Sincerely